# Factors associated with mechanical device-related complications in tube fed patients: A multicenter prospective cohort study

**Fernanda Raphael Escobar Gimenes**[1]*, **Flávia Fernanda Luchetti Rodrigues Baracioli**[2], **Adriane Pinto de Medeiros**[1], **Patricia Rezende do Prado**[3], **Janine Koepp**[4], **Marta Cristiane Alves Pereira**[1], **Camila Baungartner Travisani**[1], **Soraia Assad Nasbine Rabeh**[1], **Fabiana Bolela de Souza**[1], **Adriana Inocenti Miasso**[1]

1 University of São Paulo at Ribeirão Preto College of Nursing, Ribeirão Preto, São Paulo, Brazil, 2 Clinics Hospital, Faculty of Medicine of Ribeirão Preto, University of São Paulo, Ribeirão Preto, São Paulo, Brazil, 3 Federal University of Acre, Acre, Brazil, 4 University of Santa Cruz do Sul, Rio Grande do Sul, Brazil

* fregimenes@eerp.usp.br

**Data Availability Statement:** All relevant data are within the manuscript and its Supporting Information files.

## Abstract

### Aims

To identify the types of nasogastric/nasoenteric tube (NGT/NET)-related adverse events and to analyze the degree of harm and the factors associated with mechanical device-related complications.

### Materials and methods

A prospective cohort study was conducted from October 2017 to April 2019 in seven Brazilian hospitals. Data from 447 adult patients with NGT/NET were collected through electronic forms. Three methods were used to assess the NGT/NET-related adverse events: (1) encouraging spontaneous reports; (2) regular visits to the wards; and (3) review of medical records. The events were classified as mechanical device-related complications and other events. The degree of harm was classified according to the World Health Organization's International Classification for Patient Safety. Data were analyzed using the R program, version 3.5.3. The following tests were applied to identify associations between the explanatory and response variables: Cochran-Armitage Chi-Square test, Fisher's exact test, and Linear-by-linear Chi-Square test. Logistic regression analysis was performed to verify the predictors of mechanical device-related complications. All analyses were performed considering a 5% significance level.

### Results

191 NGT/NET-related adverse events were identified in 116 patients; the majority were mechanical device-related complications and resulted in mild harm to the patient. At the moment of the event, patients had a mean of 3.27 comorbidities, were highly dependent on nursing care, with high risk of death and altered level of consciousness. There was no association between the degree of harm and the care complexity, disease severity or level of

**Funding:** FREG received the 2019 Sigma Small Grants (Recipient ID number: 16230) from Sigma Theta Tau International. The funder had no role in study design, data collection and analysis, decision to publish, or preparation of the manuscript.

**Competing interests:** The authors have declared that no competing interests exist.

consciousness. Intensive care was the strongest predictor for mechanical device-related complications and critical patients had a four times greater likelihood of presenting this type of event when compared to patients receiving minimal care.

## Conclusion

Intensive care patients should receive special attention as the complexity of care is an important predictor for mechanical device-related complications in tube fed patients.

## Introduction

Feeding tubes are common in acute and chronic care settings for the delivery of enteral nutrition and/or medications to patients of all ages [1]. These enteral access devices may be used long term to provide nutrients through extended periods of medical need and lifelong (i.e., gastrostomy), or short term to provide nutrients for optimal functioning through periods of illness, trauma, or arduous medical therapies (i.e., nasogastric tube–NGT and nasoenteric tube —NET) [2]. In hospital settings, NGT/NETs are commonly used for temporary enteral nutrition support as they can be inserted at the bedside by trained clinicians [2]. However, nurses usually perform the procedure blindly at the patient's bedside, with this practice possibly causing serious and fatal adverse events (AEs) [3, 4]. For example, 51 reports of pneumothorax following feeding tube placement were reported from January 2012 to July 2017 in the U.S.A. In most cases, there was a need for urgent intervention, including decompression with a needle or insertion of a chest drain. Some of these events were associated with cardiac arrest and death [5].

Other complications following NGT/NET insertion include misplacement, displacement, bronchoaspiration, epistaxis, perforation of internal organs, nasal pressure injury, tube obstruction and inadvertent feeding tube removal [6, 7]. These are called mechanical device-related complication and are associated with poor patient outcomes [2], especially in critical care patients, with several comorbidities and a reduced level of consciousness [8]. In Brazil, severe and fatal NGT/NET-related AEs have been reported in the media, being mainly caused by tubing misconnections, which resulted in the infusion of enteral nutrition into the vein, followed by patient death [9].

Although NGT/NET-related AEs are relatively common in hospital settings, with significant morbidity and mortality, the issue has not been extensively studied, especially in low- and middle-income countries [10], including Brazil. Studies conducted in this area can reduce that gap and improve the care provided to patients with short-term enteral access devices.

This is the first study on NGT/NET-related AEs carried out on a large scale in Latin America and was conducted to identify the types of NGT/NET-related AEs and to analyze the degree of harm caused to patients and the factors associated with the mechanical device-related complications.

## Materials and methods

### Study design

This article is part of a broader research project on NGT/NET-related AEs [10]. This was a multicenter prospective cohort study.

## Setting

Seven centers across Brazil participated in this study, including a mix of community and university, public and private hospitals. Most hospitals were high complexity care centers and included: Hospital das Clínicas do Acre (HCA), Hospital Geral de Fortaleza (HGF), Hospital das Clínicas da Faculdade de Medicina de Ribeirão Preto da Universidade de São Paulo (HCFMRP-USP), Hospital Estadual Américo Brasiliense (HEAB), Hospital Estadual Sumaré (HES), Hospital São Vicente de Paulo (HSVP) and Hospital Santa Cruz do Rio Grande do Sul (HSCRGS). The clinical medical ward was chosen for data collection as it aggregates adult patients from different medical specialties, with a high number of tube fed patients.

## Participants

The study population was composed of patients that required a short-term feeding tube during the hospital stay. Inclusion criteria were: to be a patient over 18 years of age; admitted to a medical ward with an NGT/NET, or a patient that required the insertion of an NGT/NET; and inpatient stay of at least 24 hours. Patients that were readmitted during the data collection period were only included in the study once.

The sample size was determined by stratified random sampling with proportional allocation by strata, where each stratum was formed by the units/wards of each hospital. Adopting the parameters of relative error of 20%, significance level of 5% and the total population of 6,564, a total sample size of 391 patients was calculated. Therefore, the sample size can be considered representative of a larger population.

The observations exceeded the required sample size, totaling 447 patients. The majority were men (233; 52.1%), white (292; 65.4%), married (292; 65.3%), with incomplete primary education (164; 36.7%), retired and/or welfare beneficiaries (184; 41.2%), and residents of the state of São Paulo (310; 69.4%). The mean age was 64.91 years (66.49 ± 16.49) and the mean length of stay was 17.32 days (11.25 ± 28.98). Most patients (245; 54.8%) were admitted to the medical wards during the afternoon shift and with an NGT/NET inserted (285; 63.8%). Regarding the principal medical diagnosis, most were admitted with circulatory system disease (128; 28.6%), more than 50.0% of the patients were alert, with 42.1% ($n = 188$) being highly dependent on nursing care and presenting a high risk of death (194; 43.4%) (S1 Dataset).

## Identification of participants

Every patient that met the inclusion criteria was included. Researchers visited the wards at least twice a week to identify the patients that required an NGT/NET during the hospital stay and that fulfilled the inclusion criteria proposed for the study. The research objectives were explained to the patients and the researchers requested their voluntary participation in the study. Upon consent, the participant signed a consent form. In the case of patients unable to answer for themselves as a result of being in an advanced stage of disease, the researchers requested written authorization from the legal guardian.

## Instruments

The data collection instruments consisted of three electronic forms developed by the research team and evaluated, in terms of face and content validity, by a panel of five experts. The forms were produced in the Portuguese language using the Survey Monkey® online platform. The experts were selected through the analysis of curricula existing in the Brazilian National Council for Scientific and Technological Development (CNPq) database and were invited to participate in the study. Upon acceptance, the access links to the electronic forms were made

available to the experts for evaluation. The modified electronic forms were tested with a pilot study including five patients admitted to the medical wards, from the first day of use of NGT/ NET to discharge from the medical ward [10].

The first electronic data collection form was used to collect data from the patient on admission and included demographic data (e.g., date of birth; city / state of origin; gender; race; marital status; education level; and profession), clinical data (e.g., principal and secondary diagnoses, according to the International Classification of Diseases [ICD], 10<sup>th</sup> edition; comorbidities; the final score of the Patient Classification System; and level of consciousness), and therapeutic data (e.g., data related to the enteral access device and enteral nutrition; methods used to confirm feeding tube placement and the results of the methods used) (S1 File).

The second electronic data collection form included variables related to the NGT/NET-related AEs (e.g., date and time of the event; type of NGT/NET-related AE and the degree of harm) (S2 File).

The third electronic form included variables related to the removal of the NGT/NET and patient follow-up (i.e., date of tube removal and the main reason; date and time of patient discharge; reason for patient discharge) (S3 File).

## Data collection

Data were collected from October 2017 to April 2019. In each center participating in the study, a regional coordinator was appointed in order to ensure the quality and comprehensiveness of the data collected. For the data collection, a mobile device (cell phone or tablet) was used, from the first day of use of the NGT/NET in the ward (or from the first day of hospitalization if the patient was admitted to the ward with the tube) until discharge from the ward (due to death or non-death).

Three methods were used to assess the NGT/NET-related AEs: *Encouraging spontaneous reports*: healthcare providers and patients/caregivers were asked to report any NGT/NET-related AEs to the researchers; *Regular visits to the wards*: at least twice a week, the researchers visited the wards, to request information about the AEs, from the healthcare providers and patients/caregivers; *Review of medical records*: researchers reviewed medical records at least twice a week to obtain information about NGT/NET-related AEs [10].

An NGT/NET-related AE was defined as any undesirable experience associated with the use of an enteral access device during the hospital stay and was classified as a mechanical device-related complication (bronchoaspiration, pneumothorax, epistaxis/nosebleed, perforation of internal organs, tube migration/displacement, nasal pressure injury, tube obstruction, inadvertent feeding tube removal, and multiple attempts to introduce the tube) [6] or another event (tubing misconnection and tube quality).

The degree of harm was assessed based on the information contained in the patient's medical record and from the patient / caregiver / healthcare provider reports. They were classified according to the World Health Organization (WHO)'s International Classification for Patient Safety [11]: None—patient outcome is not symptomatic or no symptoms detected and no treatment is required; Mild—patient outcome is symptomatic, symptoms are mild, loss of function or harm is minimal or intermediate but short term, and no or minimal intervention (e.g., extra observation, investigation, review or minor treatment) is required; Moderate— patient outcome is symptomatic, requiring intervention (e.g., additional therapeutic treatment), an increased length of stay, or causing permanent or long term harm or loss of function; Severe—patient outcome is symptomatic, requiring life-saving intervention or major surgical/medical intervention, shortening life expectancy or causing major permanent or long term harm or loss of function; Death–on balance of probabilities, death was caused or brought forward in the short term by the incident.

The level of consciousness was verified by the research team using the ACDU Scale, which corresponds to: Alert, Confused, Drowsy and Unconscious [12]. It is a simple, quick and useful scale to assess, at the bedside, the patient's level of consciousness. It is often used by nurses and other healthcare providers in different contexts. This scale has also proved to be superior to the others in the early identification of neurological deterioration in critically ill patients on wards [13, 14].

The care complexity of the patients was assessed at hospital admission by trained nurses, members of the research team. For this, the Patient Classification System (PCS) [15] was used, as recommended by the Federal Nursing Council of Brazil [16]. The instrument was developed with the aim of classifying patients according to the degree of dependence on the nursing team. The instrument has nine critical indicators: mental status, oxygenation, vital signs, mobility, walking, feeding, body care, elimination and therapy. Accordingly, the points are divided into five categories that correspond to the care complexity: minimal care (score 9 to 14), intermediate care (score 15 to 20), high dependence care (score 21 to 26), semi-intensive care (score 27 to 31) and intensive care (score >31).

Disease severity was assessed from the patient medical records using the Charlson Comorbidity Index (CCI) [17]. The original study was conducted from 20 clinical conditions empirically selected based on the effect of these conditions on the prognosis of hospitalized patients [17]. The CCI consists of a method for categorizing patient comorbidities, according to the ICD-10. The aim of the CCI is to measure the severity of the patient, regardless of the main diagnosis, that is, to assess the prediction of the risk of death. The final score is the result of the sum of the weights attributed to the comorbidities registered as secondary diagnoses, with higher scores indicating a higher risk of patient death. Based on the final CCI score, the patients were stratified into three groups: low risk (score 1 to 2); moderate risk (score 3 to 4) and high risk (score ≥5) [18].

## Data analysis

Data were transferred from the Survey Monkey® online platform to Microsoft Excel® Program spreadsheets. In the descriptive statistical analysis, the calculation of proportions and measures of central tendency and variability were performed. To verify the presence of an association between the explanatory variables "care complexity", according to PCS [15]; "disease severity", measured by the CCI [17]; and "level of consciousness", assessed by the ACDU scale [12], with the response variable "mechanical device-related complications" (yes/no), the Cochran-Armitage chi-square test [19] was used.

To verify the presence of an association between the explanatory variables "use of invasive breathing device" at the time of the event (yes / no) with the response variable "mechanical device-related complications" (yes / no), Fisher's exact test was used.

Crosses were also performed between the explanatory variables "care complexity", "disease severity" and "level of consciousness" at the time of the event and the response variable "degree of harm". In this case, considering that both variables are ordinal, the linear-by-linear chi-square test was applied [20].

Logistic regression analysis was carried out to verify the predictors of mechanical device-related complications in tube fed patients. The explanatory variables used in the analyses were: "care complexity" (minimum care, intermediate care, high dependency care, semi-intensive care and intensive care); "disease severity" (no risk, low risk, moderate risk and high risk); "level of consciousness" (alert, confused, drowsy or unconscious); "use of invasive breathing device" (yes/no); "age" (years); "length of stay" (months); "length of tube use" (months) and "reason for discharge" (death/non-death).

Finally, to perform the logistic regression analysis, the response variable was the occurrence of mechanical device-related complications (yes/no) during the hospital stay. The selection of explanatory variables for the final model was performed using the Likelihood Ratio test. The Variance Inflation Factor (VIF) was also used to identify the presence of multicollinearity between the explanatory variables of the model, that is, to assess the existence of a strong association between two or more explanatory variables of the logistic regression model [21]. Values above 5 were indicative of multicollinearity. All analyses were performed using the R program, version 3.5.3 considering a significance level of 5% ($\alpha = 0.05$).

## Ethical considerations

The study was approved by the Research Ethics Committee of the University of São Paulo at Ribeirão Preto College of Nursing (CAAE: 56166016.3.1001.5393), according to Resolution No. 466/2012, of the National Council of Research Ethics of the Ministry of Health, which addresses ethics in research with human subjects. Informed consent was obtained from each patient, or their guardian, prior to inclusion in the study.

## Results

A total of 191 NGT/NET-related AEs were identified in 116 patients (25.9%), expressly, a mean of 1.64 events/patient (1 ± 0.98), with a minimum of one and a maximum of six. Most NGT/NET-related AEs were mechanical device-related complications (185; 96.8%) and resulted in mild harm to the patient (95; 49.7%); however, in 15.7% (n = 30) of the cases harm was severe or moderate.

The most frequent mechanical device-related complication was inadvertent feeding tube removal (134; 70.1%). Although infrequent, there were two cases of tubing misconnection that resulted in moderate harm to the patients. No cases of pneumothorax or perforation of internal organs were identified (Table 1).

At the time of the event, patients presented a mean of 3.27 comorbidities (3 ± 2.15), were highly dependent on nursing care (95; 49.7%), with a high risk of death (80; 41.9%), with altered level of consciousness (101; 52.9%) and without an invasive breathing device (154; 80.6%) (Table 2).

Table 3 shows that there was no association between the degree of harm and care complexity (p = .997), disease severity (p = .794) or level of consciousness (p = .909).

Table 4 shows that there was an association between mechanical device-related complication and disease severity (p = .041) or level of consciousness (p = .041). Furthermore, in most cases, patients were at high risk of death (87; 12.1%), alert (117; 16.3%) and did not have an invasive breathing device (162; 22.6%).

A logistic regression model was developed to assess the predictors of mechanical device-related complications in tube fed patients. Variables with values of $p \leq .05$ were included in the model. The selection of the explanatory variables was performed through the Likelihood Ratio and VIF test.

Table 5 shows that care complexity, level of consciousness and reason for discharge (death/non-death) had a significant contribution to the model.

The results of the gross OR analysis show that, when compared to patients that required minimal nursing care, patients classified as high dependence care and intermediate care were more likely to have mechanical device-related complications. However, in the adjusted analysis, semi-intensive care was not a predictor of the event. Intensive care was the strongest predictor for mechanical device-related complications (OR = 4.72; 95%CI: 1.43–15.52; p = .011).

**Table 1. Type of NGT/NET-related AE and degree of harm (N = 191).**

| Variables | n | % |
|---|---|---|
| AE—median ± SD (min—max) | 1 ± 0.98 (1–6) | - |
| **Mechanical AE** | | |
| Nasal pressure injury | 1 | 0.5 |
| Epistaxis/nosebleed | 5 | 2.6 |
| Multiple attempts to introduce the tube | 5 | 2.6 |
| Migration/displacement | 6 | 3.1 |
| Bronchoaspiration | 9 | 4.7 |
| Tube obstruction | 25 | 13.1 |
| Inadvertent tube removal | 134 | 70.1 |
| Total | 185 | 96.9 |
| **Other AE** | | |
| Tubing misconnection | 2 | 1.0 |
| Tube quality | 4 | 2.0 |
| Total | 6 | 3.1 |
| **Degree of harm**[a] | | |
| Death | 0 | 0.0 |
| Severe | 3 | 1.6 |
| Moderate | 27 | 14.1 |
| None | 64 | 33.5 |
| Mild | 95 | 49.7 |
| Total | 189 | 98.9 |

AE, Adverse event; SD, Standard deviation.

[a]Counts do not sum to the total of 191 due to missing values.

Patients receiving intensive care were four times more likely to present mechanical device-related complications when compared to patients receiving minimal care.

Changes in the level of consciousness reduced the likelihood of the tube fed patient presenting mechanical device-related complications, with drowsiness reducing this likelihood by approximately 75.0% (OR = 0.24; 95%CI: 0.10–0.61; $p$ = .003) (Table 6).

## Discussion

Feeding tubes are indicated for patients that have a functional and accessible gastrointestinal tract, however, are unable to consume or absorb enough nutrients to sustain adequate nutrition and hydration [2]. Although widely used in hospitals, long-term care and home settings, these enteral access devices are not exempt from serious and potentially life-threatened adverse events.

In this study, NGT/NET-related AEs were identified in 25.9% of the patients. This result is higher than that found in a previous study carried out by Michel and colleagues (15.4%) [22]. This study, however, included patients admitted to the medical, surgical and obstetric wards, characterizing a different patient profile.

Most of the NGT/NET-related AEs caused mild harm to the patient, although severe or moderate harm occurred in 15.7% of the cases. Studies have shown that patients using NGT/NET are subject to AEs in the hospital setting [3, 4, 23–25], however, none have aimed to assess the degree of harm. This is the first study to do so with this patient profile.

**Table 2. Distribution of variables of the patients with NGT/NET-related AE (N = 191).**

| Variables | n | %[a] |
|---|---|---|
| No. of comorbidities—median ± *SD* (min—max) | 3 ± 2.15 (0–9) | - |
| **Care complexity (PCS)** | | |
| Intensive | 8 | 4.2 |
| Minimum | 18 | 9.4 |
| Intermediate | 30 | 15.7 |
| Semi-intensive | 36 | 18.8 |
| High dependency | 95 | 49.7 |
| Total | 187 | 97.9 |
| **Disease severity (CCI)** | | |
| No risk | 16 | 8.4 |
| Low risk | 40 | 20.9 |
| Moderate risk | 55 | 28.8 |
| High risk | 80 | 41.9 |
| Total | 191 | 100.0 |
| **Level of consciousness (ACDU scale)** | | |
| Drowsy | 10 | 5.2 |
| Unconscious | 17 | 8.9 |
| Confused | 74 | 38.7 |
| Alert | 88 | 46.1 |
| Total | 189 | 98.9 |
| **Use of invasive breathing device** | | |
| Yes | 36 | 18.8 |
| No | 154 | 80.6 |
| Total | 190 | 99.5 |

*SD*, Standard deviation; PCS, Patient Classification System; CCI, Charlson Comorbidity Index; ACDU, Alert, Confused, Drowsy and Unconscious.

[a]Counts do not sum to the total of 191 due to missing values.

A report was published by NHS Improvement [26] on Never Events (defined as serious, largely preventable patient safety incidents that should not occur if healthcare providers have implemented existing national guidance or safety recommendations) that occurred between 1st April 2018 and 31st January 2019. Of the 423 serious incidents reported, NGT or orogastric tube-related AEs accounted for 6.4% and were related to misplacement of tubes in the respiratory tract with food administered to the patient.

Hospital mortality and length of stay, two important hospital quality indicators, can be increased by NGT/NET-related AEs [27]. Therefore, to provide patients with safe, high quality, and compassionate care, a systematic approach should be in place [7], including optimal communication and standardization across all steps of the NGT/NET use [28], development of and adherence to policies and standardized procedures for daily practice and decision-making related to patient care [7]; a whole-system approach to help prevent misplacement of NGT/NET; education and training; formal monitoring; and reporting systems to improve patient safety [29].

Mechanical device-related complications were the most frequent (96.8%) NGT/NET-related AE, with inadvertent feeding tube removal being the recurrent type (70.1%), corroborating the results of previous studies [4, 23]. This rate is much higher than recommended by the International Life Sciences Institute (ILSI-Brazil), which proposes a target of <10% on

**Table 3. Analysis of the association between the degree of harm and care complexity, disease severity and level of consciousness (N = 191).**

| Variables | Degree of harm | | | | | | | | | | P-value |
|---|---|---|---|---|---|---|---|---|---|---|---|
| | None | | Low risk | | Moderate risk | | High risk | | Total[a] | | |
| | n | % | n | % | n | % | n | % | n | % | |
| **Care complexity (PCS)** | | | | | | | | | | | |
| Intensive | 1 | 0.5 | 4 | 2.1 | 3 | 1.6 | 0 | 0.0 | 8 | 4.2 | .997[b] |
| Minimum | 5 | 2.6 | 10 | 5.2 | 12 | 6.3 | 0 | 0.0 | 17 | 8.9 | |
| Intermediate | 8 | 4.2 | 15 | 7.8 | 6 | 3.1 | 0 | 0.0 | 29 | 15.2 | |
| Semi-intensive | 12 | 6.3 | 17 | 8.9 | 5 | 2.6 | 2 | 1.0 | 36 | 18.8 | |
| High dependency | 37 | 19.4 | 45 | 23.6 | 11 | 5.7 | 1 | 0.5 | 94 | 49.2 | |
| Total | 63 | 33.0 | 91 | 47.6 | 27 | 14.1 | 3 | 1.6 | 184 | 96.3 | |
| **Disease severity (CCI)** | | | | | | | | | | | |
| No risk | 5 | 2.6 | 11 | 5.7 | 0 | 0.0 | 0 | 0.0 | 16 | 8.4 | .794[b] |
| Low risk | 15 | 7.8 | 16 | 8.8 | 7 | 3.7 | 0 | 0.0 | 38 | 19.9 | |
| Moderate risk | 19 | 9.9 | 28 | 14.6 | 7 | 3.7 | 1 | 0.5 | 55 | 28.8 | |
| High risk | 25 | 13.1 | 39 | 20.4 | 13 | 6.8 | 2 | 1.0 | 79 | 41.4 | |
| Total | 64 | 33.5 | 94 | 49.2 | 27 | 14.1 | 3 | 1.6 | 188 | 98.4 | |
| **Level of consciousness (ACDU scale)** | | | | | | | | | | | |
| Drowsy | 1 | 0.5 | 2 | 1.0 | 6 | 3.1 | 1 | 0.5 | 10 | 5.2 | .909[b] |
| Confused | 16 | 8.4 | 42 | 22.0 | 14 | 7.3 | 1 | 0.5 | 73 | 38.2 | |
| Unconscious | 10 | 5.2 | 5 | 2.6 | 1 | 0.5 | 1 | 0.5 | 17 | 8.9 | |
| Alert | 37 | 19.4 | 43 | 22.5 | 6 | 3.1 | 0 | 0.0 | 86 | 45.0 | |
| Total | 64 | 33.5 | 92 | 48.2 | 27 | 14.1 | 3 | 1.6 | 186 | 97.4 | |

PCS, Patient Classification System; CCI, Charlson Comorbidity Index; ACDU, Alert, Confused, Drowsy and Unconscious.

[a]Counts do not sum to the total of 191 due to missing values.

[b]Linear-by-linear chi-square test.

medical wards [30]. Studies have identified the causes of inadvertent feeding tube removal on clinical wards, which included withdrawal by the patients themselves due to delirium [31]. In the present study, most patients were highly dependent on nursing care, with high risk of death and with a mean of 3.27 comorbidities. Furthermore, 38.7% were confused at the time of the AE.

Another common cause of inadvertent feeding tube removal is tube obstruction [7, 32, 33]. In this study, this event was responsible for 13.1% of all NGT/NET-related AEs. According to the literature, the rates related to tube obstruction vary from 12.5 to 45.0%. The event may be caused by mixing different medications together, failure to crush simple compressed tablets to a fine powder, failure to administer each medication separately, failure to wash the tube with at least 15 ml of water before and after medication administration, and failure to pause the enteral nutrition during medication administration [7, 33, 34]. Some interventions are proposed to reduce the risk of tube obstruction and include: policies and procedures to ensure safe practices by healthcare teams; review by a pharmacist of each medication order to determine whether the enterally administered medication will be safe; and the institution and following of nursing policies and procedures to safely prepare and administer medications [7].

Although uncommon, bronchoaspiration was identified in this study (4.7%), with the event possibly being related to failures to confirm proper feeding tube placement [24, 25, 35]. Bronchoaspiration is an underreported mechanical device-related complication, most likely due to the difficulty in establishing a medical diagnosis. This event can progress to aspiration

**Table 4. Analysis of the association between mechanical device-related complication and care complexity, disease severity, level of consciousness and use of an invasive breathing device (N = 717).**

| Variables | Mechanical device-related complication | | | | | | P-value |
| --- | --- | --- | --- | --- | --- | --- | --- |
| | Yes | | No | | Total[a] | | |
| | n | % | n | % | n | % | |
| **Care complexity (PCS)** | | | | | | | .862[b] |
| Intensive | 7 | 1.0 | 17 | 2.4 | 24 | 3.3 | |
| Minimum | 14 | 2.0 | 78 | 10.9 | 92 | 12.8 | |
| Semi-intensive | 24 | 3.3 | 108 | 15.1 | 132 | 18.4 | |
| Intermediate | 44 | 6.1 | 91 | 12.7 | 135 | 18.8 | |
| High dependency | 89 | 12.4 | 215 | 30.0 | 304 | 42.4 | |
| Total | 178 | 24.8 | 509 | 71.0 | 687 | 95.8 | |
| **Disease severity (CCI)** | | | | | | | .041[b] |
| No risk | 13 | 1.8 | 29 | 4.0 | 42 | 5.8 | |
| Low risk | 38 | 5.3 | 83 | 11.6 | 121 | 16.9 | |
| Moderate risk | 52 | 7.3 | 123 | 17.1 | 175 | 24.4 | |
| High risk | 87 | 12.1 | 289 | 40.3 | 376 | 52.4 | |
| Total | 190 | 26.5 | 529 | 73.8 | 714 | 99.6 | |
| **Level of consciousness (ACDU scale)** | | | | | | | .041[b] |
| Drowsy | 6 | 0.8 | 50 | 7.0 | 56 | 7.8 | |
| Unconscious | 17 | 2.4 | 64 | 8.9 | 81 | 11.3 | |
| Confused | 48 | 6.7 | 111 | 15.5 | 159 | 22.2 | |
| Alert | 117 | 16.3 | 300 | 41.8 | 417 | 58.1 | |
| Total | 188 | 26.2 | 525 | 73.2 | 713 | 99.4 | |
| **Use of invasive breathing device** | | | | | | | .967[c] |
| Yes | 28 | 3.9 | 91 | 12.7 | 119 | 16.6 | |
| No | 162 | 22.6 | 435 | 60.7 | 597 | 83.3 | |
| Total | 190 | 26.5 | 526 | 73.4 | 716 | 99.9 | |

PCS, Patient Classification System; CCI, Charlson Comorbidity Index; ACDU, Alert, Confused, Drowsy and Unconscious.

[a]Counts do not sum to the total of 717 due to missing values.

[b]Cochran-Armitage Chi-Square test.

[c]Fisher's exact test.

pneumonia, cardiac arrest, neurological sequelae and death [36], with bronchoaspiration only confirmed by anatomopathological exams in most cases.

Researchers have proposed a method for diagnosing bronchoaspiration by collecting samples of secretions from the airways in order to analyze the presence of substances from the gastrointestinal tract [36]. They analyzed the secretions obtained during the lower airway aspiration procedure in critically ill patients and found that, of the 8,857 samples collected, 31.3% indicated the presence of pepsin. Despite the evidence of bronchoaspiration, less than 1.0% of the patients showed signs and symptoms related to the phenomenon [36].

There are no studies proposing a measure of association that confirms the direct relationship between exposure and effect. Therefore, it is not possible to state that bronchoaspiration due to NGT/NET occurred in this study, considering the existence of other confounding variables such as age, altered level of consciousness and the use of an invasive breathing device.

Patients may be at increased risk for aspiration due to a number of factors, including inability to protect the airway, presence of an enteral access device, mechanical ventilation, age >70 years, reduced level of consciousness, poor oral care, inadequate nurse-to-patient ratio, supine

**Table 5. Independent variables included in the final logistic regression model for the analysis of predictors of mechanical device-related complications in tube fed patients.**

| Variables | Estimate | SD | Z-value | Pr(>\|z\|) |
|---|---|---|---|---|
| (Intercept) | -1.1215 | 0.421 | -2.661 | 0.007 |
| Age | -0.0104 | 0.005 | -1.837 | 0.066 |
| **Care complexity (PCS)** | | | | |
| High dependency care | 1.3974 | 0.348 | 4.007 | <0.001 |
| Intensive care | 1.6369 | 0.613 | 2.667 | 0.007 |
| Intermediate care | 1.2647 | 0.364 | 3.469 | <0.001 |
| Semi-intensive care | 0.5240 | 0.404 | 1.295 | 0.195 |
| **Level of consciousness (ACDU scale)** | | | | |
| Confused | -0.2039 | 0.233 | -0.873 | 0.382 |
| Unconscious | -0.2497 | 0.340 | -0.734 | 0.462 |
| Drowsy | -1.3978 | 0.471 | -2.965 | 0.003 |
| **Reason for Discharge** | | | | |
| Death | -0.6221 | 0.249 | -2.489 | 0.012 |

*SD*, Standard Deviation; PCS, Patient Classification System; ACDU, Alert, Confused, Drowsy and Unconscious.

positioning, neurologic deficits, gastroesophageal reflux, transport out of the ICU and the use of bolus intermittent enteral nutrition [37]. Actions for the prevention of bronchoaspiration must be implemented in health services to improve patient safety, including [2, 7]: identifying and monitoring patients at high risk for aspiration; keeping the tube in a post-pyloric position in critically ill patients at high risk of aspiration; keeping the head of the bed elevated between 30 and 45° and changing the position at least every 4 hours (if this action is contraindicated, the reverse-Trendelenburg position is recommended); administering enteral nutrition by means of a continuous infusion pump in patients at high risk of aspiration and with delayed gastric emptying; measuring gastric residual volume in critically ill patients, at 4-hour intervals; in the case of volumes between 250 and 500 mL, considering the implementation of

**Table 6. Model of logistic regression analysis with the response variable mechanical device-related complication (N = 197).**

| Variables | Gross OR (95%CI) | P-value | Adjusted OR (95%CI) | P-value[a] |
|---|---|---|---|---|
| **Care complexity (PCS)** | | | | |
| Minimum care (reference) | | | | |
| High dependency care | 2.64 (1.39–5.02) | .003 | 3.56 (1.83–6.93) | .001 |
| Intensive care | 1.98 (0.66–5.97) | .224 | 4.72 (1.43–15.52) | .011 |
| Intermediate care | 2.74 (1.37–5.49) | .004 | 3.23 (1.60–6.54) | .001 |
| Semi-intensive care | 1.26 (0.60–2.67) | .545 | 1.60 (0.72–3.52) | .247 |
| **Level of consciousness (ACDU scale)** | | | | |
| Alert (reference) | | | | |
| Confused | 0.98 (0.63–1.51) | .918 | 0.81 (0.52–1.28) | .371 |
| Unconscious | 0.61 (0.34–1.1) | .103 | 0.76 (0.39–1.48) | .423 |
| Drowsy | 0.32 (0.13–0.78) | .012 | 0.24 (0.10–0.61) | .003 |
| **Reason for Discharge** | | | | |
| Death vs. Non-Death | 0.59 (0.37–0.93) | .023 | 0.51 (0.31–0.83) | .007 |

OR, Odds Ratio; 95%CI, 95% Confidence Interval; PCS, Patient Classification System; ACDU, Alert, Confused, Drowsy and Unconscious.

[a]*p* values were calculated using the Likelihood Ratio test.

measures aimed at reducing the risk of aspiration; checking tube placement and recording the initial external measurement every 4 hours; checking tube placement after episodes of vomiting, coughing and after tube dislodgement; monitoring gastrointestinal activity changes that include presence of nausea, vomiting, reflux, feeling of fullness, distension, absence of bowel sounds, abdominal discomfort, pain or cramps, at 4-hour intervals; maintaining minimal doses of sedation; and considering the use of prokinetic agents (such as metoclopramide), when clinically feasible, in patients at high risk for aspiration.

Although infrequent, there were two cases of tubing misconnection in this study, which resulted in moderate harm to the patient. Serious and fatal AEs related to feeding tube misconnection have been reported [3, 9], resulting in additional treatment, prolonged hospitalization, disability or death and, for these reasons, patients should be continuously monitored for risk [26]. One of the main reasons for tubing misconnections is that many types of tubing for different types of medical devices use luer fittings. These connectors allow functionally dissimilar tubes or catheters to be connected together. Therefore, reducing the likelihood of this safety incident includes equipment design solutions that prevent the user from making a misconnection or prompt the user to make the correct connection, as well as policies that support providers [38].

There was an association between mechanical device-related complications and disease severity and level of consciousness. This result is aligned with previous research that also identified age, principal medical diagnosis, and use of an invasive breathing device as risk factors for NGT/NET-related AEs [39]. The altered level of consciousness may be related to the presence of delirium, characterized by acute changes in consciousness, accompanied by inattention and changes in cognition or perceptual disturbances [40]. The relationship between the number of comorbidities and delirium was previously established. Patients with multiple diseases are usually exposed to polypharmacy, which is a risk factor for delirium, especially in older adults [41]. Other factors include pre-existing dementia, high blood pressure, alcoholism and the disease burden. Delirium is also considered a risk factor for early death, in addition to increasing the length of stay and overall hospital costs [40]. Despite the importance of the early recognition of delirium, research conducted in a Brazilian ICU found that nurses have doubts about the signs and symptoms of delirium and the management of critical patients. Furthermore, nurses are unaware of the delirium screening instruments [42]. Nurses must be able to recognize patients at risk for the development of acute confusion, through the application of reliable instruments. Early diagnosis and treatment of delirium improve patient outcomes and help reduce AEs caused by this clinical condition [40], especially in tube fed patients.

There was no association between the degree of harm and the patient care complexity, disease severity and level of consciousness. However, among patients that suffered severe harm, none were alert and most of those that suffered moderate harm were confused. Healthcare providers should be aware of the risks and monitor level of consciousness continually in order to improve outcomes and the quality of care provided to patients with NGT/NET.

In the logistic regression analysis, only the variables care complexity, level of consciousness and reason for discharge (death/non-death) were predictors of mechanical device-related AEs, with intensive care being the strongest predictor of the event. The literature confirmed the negative influence of these variables on general AEs in critically ill patients, however, other studies that have performed similar analyses with patients with NGT/NET were not identified in the literature. Accordingly, future research aimed at this patient profile is recommended in order to explain the causes of this important phenomenon [43].

In the adjusted logistic regression analysis, it was observed that altered level of consciousness decreased the chances of the patient with NGT/NET presenting a mechanical device-

related complication, with drowsiness reducing this risk by about 75%. These results differ from other studies, possibly due to the differences in the methodologies adopted [23, 41].

Drowsy patients present a state of torpor, with significant impairment of consciousness, however, they can be awakened by painful stimuli. Therefore, the ability of these patients to remove the tube may be limited, justifying the result found.

Successful management of these risks depends on nurses developing competencies related to the management of health services. These competencies are determined by the professional's level of knowledge, technical and non-technical skills and a proactive attitude. The implications of managerial competencies refer to the mobilization, integration and transfer of knowledge and skills to add economic value to the institution and give social value to the individuals [44]. Managerial competencies can be described as leadership, safety culture, teamwork, communication, emotional intelligence and decision-making in risk management and planning. Considering issues related to patient safety, the nurse's managerial skills include constant concern with AEs, involvement of the patient and caregiver in the care process, and the identification and reduction of NGT/NET-related AEs. The learning and development of these competencies should start from the training of the nurse, including the discussion of the aspects that permeate patient safety in the undergraduate nursing curricula [43].

## Limitations and strengths

Although the results made it possible to identify NGT/NET-related AEs, it is important to consider the aspects that may have limited the study. First, the identification of NGT/NET-related AEs was performed based on patients/caregivers' and the health team's spontaneous reports and through reviewing medical records. Therefore, the results presented here may not portray the magnitude of the problem due to underreporting and flaws in the records. Despite this, the results have the potential to contribute to the management of care through the knowledge of the most frequent NGT/NET-related AEs and the predictors of mechanical device-related AEs in tube fed patients.

Second, we did not account for the magnitude of differences between hospitals. There are no studies reporting NGT/NET-related AEs across multiple hospital sites at a national level. Therefore, a multicentre study may ascertain better generalisability of the data. A comparative analysis between facilities is recommended in future studies.

## Conclusion

In this cohort, NGT/NET-related AEs were common, with the mechanical device-related AE being the most frequent type in tube fed patients, resulting in mild harm. The strongest predictor for mechanical device-related AE was intensive care, with these patients having a four times greater chance of presenting this type of event when compared to patients receiving minimal care. Intensive care patients should receive special attention, due to the care complexity being an important predictor of mechanical device-related complications in tube fed patients.

## Supporting information

**S1 Dataset. Patient and adverse event data.**
(XLSX)

**S1 File. Patient admission form.**
(PDF)

**S2 File. NGT/NET-related adverse events form.**
(PDF)

**S3 File. Patient follow-up form.**
(PDF)

## Acknowledgments

We thank the patients and caregivers that trusted in our work and agreed to participate in the study. We also thank the health teams of the hospitals that collaborated for the study to be carried out. We also thank Rhanna Emanuela Fontenele Lima de Carvalho, Thalyta Cardoso Alux Teixeira and Ligia Menezes de Freitas for assistance with data collection.

## Author Contributions

**Conceptualization:** Fernanda Raphael Escobar Gimenes.

**Data curation:** Fernanda Raphael Escobar Gimenes, Flávia Fernanda Luchetti Rodrigues Baracioli, Marta Cristiane Alves Pereira.

**Formal analysis:** Fernanda Raphael Escobar Gimenes, Flávia Fernanda Luchetti Rodrigues Baracioli, Camila Baungartner Travisani, Soraia Assad Nasbine Rabeh.

**Funding acquisition:** Fernanda Raphael Escobar Gimenes.

**Investigation:** Fernanda Raphael Escobar Gimenes, Adriane Pinto de Medeiros, Patricia Rezende do Prado, Janine Koepp, Marta Cristiane Alves Pereira, Camila Baungartner Travisani, Fabiana Bolela de Souza, Adriana Inocenti Miasso.

**Methodology:** Fernanda Raphael Escobar Gimenes, Patricia Rezende do Prado, Janine Koepp.

**Project administration:** Fernanda Raphael Escobar Gimenes.

**Supervision:** Fernanda Raphael Escobar Gimenes, Adriane Pinto de Medeiros, Patricia Rezende do Prado, Janine Koepp.

**Validation:** Fernanda Raphael Escobar Gimenes, Flávia Fernanda Luchetti Rodrigues Baracioli, Adriane Pinto de Medeiros, Marta Cristiane Alves Pereira, Soraia Assad Nasbine Rabeh, Fabiana Bolela de Souza, Adriana Inocenti Miasso.

**Visualization:** Fernanda Raphael Escobar Gimenes.

**Writing – original draft:** Fernanda Raphael Escobar Gimenes, Flávia Fernanda Luchetti Rodrigues Baracioli, Adriane Pinto de Medeiros, Patricia Rezende do Prado, Janine Koepp, Marta Cristiane Alves Pereira, Camila Baungartner Travisani, Soraia Assad Nasbine Rabeh, Fabiana Bolela de Souza, Adriana Inocenti Miasso.

**Writing – review & editing:** Fernanda Raphael Escobar Gimenes, Flávia Fernanda Luchetti Rodrigues Baracioli, Adriane Pinto de Medeiros, Patricia Rezende do Prado, Janine Koepp, Marta Cristiane Alves Pereira, Camila Baungartner Travisani, Soraia Assad Nasbine Rabeh, Fabiana Bolela de Souza, Adriana Inocenti Miasso.

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
