## [Decision Letter · Decision Letter 0]

24 Aug 2020

PONE-D-20-12166

Factors associated with nasogastric/nasoenteric feeding tube-related adverse events: a multicenter prospective cohort study

PLOS ONE

Dear Dr. Gimenes,

Thank you for submitting your manuscript to PLOS ONE. After careful consideration, we feel that it has merit but does not fully meet PLOS ONE’s publication criteria as it currently stands. Therefore, we invite you to submit a revised version of the manuscript that addresses the points raised during the review process.

Please response appropriately and specifically to the reviewers' comments. The decision of this work will depend on your revision. 

We look forward to receiving your revised manuscript.

Kind regards,

Chun Chieh Yeh, M.D., Ph.D.

Academic Editor

PLOS ONE

Additional Editor Comments:

Sorry for the delay in response because it is very challenging to finding appropriate reviewers in this field. The publication about this topic is surprisingly low . However, our reviewers still give some good comments about this articles. Decision will be made based on your on-time and specific responses to the comments.

2. Please include additional information regarding the data collection instruments used in the study and ensure that you have provided sufficient details that others could replicate the analyses.

For instance, if you developed a questionnaire as part of this study and it is not under a copyright more restrictive than CC-BY, please include a copy, in both the original language and English, as Supporting Information.

3. In your Methods section, please provide additional information about the participant recruitment method and the demographic details of your participants.

Please ensure you have provided sufficient details to replicate the analyses such as:

a) the recruitment date range (month and year),

b) a table of relevant demographic details and

c) a statement as to whether your sample can be considered representative of a larger population.

Reviewers' comments:

Reviewer's Responses to Questions

**Comments to the Author**

1. Is the manuscript technically sound, and do the data support the conclusions?

Reviewer #1: Yes

Reviewer #2: Yes

2. Has the statistical analysis been performed appropriately and rigorously? 

Reviewer #1: Yes

Reviewer #2: Yes

3. Have the authors made all data underlying the findings in their manuscript fully available?

Reviewer #1: Yes

Reviewer #2: Yes

4. Is the manuscript presented in an intelligible fashion and written in standard English?

Reviewer #1: Yes

Reviewer #2: Yes

5. Review Comments to the Author

Reviewer #1: The authors assessed the types of nasogastric/nasoenteric feeding tube (NGT/NET)-related adverse events, the degree of harm, and to analyze the factors associated with mechanical events.

The strength of this study is that it is a prospective and multi-center research.

However, the disadvantage of this study is that it lacks novelty.

Additionally, the authors should consider the differences between facilities.

Reviewer #2: Overall, the research is well described, and the supporting information is enough to elucidate about the frequency and risk factors of mechanical adverse effects cause by NGT/NET placement and use. Taking into the account the frequency of NGT/NET use, the number of studies on the topic is surprising low, therefore research on the topic is welcomed. The manuscript is suitable for publication in this journal subject to major revisions.

Major revisions

1.The manuscript is essentially about the occurrence and factors associated with mechanical AEs (96%) related to NGT/NET. The multivariate analysis was done to identify predictors of mechanical AEs. The entire manuscript, including the title must reflect this. For instance, the title must be changed to “Factors associated with nasogastric/nasoenteric feeding tube related mechanic complications: a multicenter prospective cohort study”. A brief description of other non-mechanical AEs (results) is sufficient.

2.Page 5, line 118 - Study design – the authors must include information on the inclusion process of the patients (consecutive ? …).

3.Page 9, 213 – “Comparisons between the means of the variables “number of comorbidities” at the time of AE 213 and “degree of harm” were performed using the Jonckheere-Terpstra test [20,21]” - The number of comorbidities is a gross measurement of comorbidities burden. The authors should explain the option of not using the CCI which is a much sensitive instrument.

4. Page 3, 64 - “can be avoided through good practices in nursing – this conclusion is not supported by the study data. The authors should be less pragmatic.

Minor revisions

1.The manuscript requires significant proof reading and revision to improve the quality of English.

2.Page 4, line 88 - define light treatment

3.Page 4, line 91 - explain how life expectancy was evaluated in relation to the AEs

4.Page 5, line 125 – “with and without residency programs” – this is not a necessary information

5. The main adverse effect was unplanned/accidental withdrawal (70.1%) – which seems to be the a major AE in non -cooperative patients, for example in stroke patients (Complications Associated With Nasogastric Tube Placement in the Acute Phase of Stroke: A Systematic Review.Nascimento A, Carvalho M, Nogueira J, Abreu P, Nzwalo H.J Neurosci Nurs. 2018 Aug;50(4):193-198. doi: 10.1097/JNN.0000000000000372.). Discussion of the implications of NGT in non-cooperative patients should be included.

6. PLOS authors have the option to publish the peer review history of their article (what does this mean?). If published, this will include your full peer review and any attached files.

Reviewer #1: No

Reviewer #2: No

---

## [Author Response · Author response to Decision Letter 0]

5 Oct 2020

We uploaded a rebuttal letter that responds to each point raised by the academic editor and reviewer(s). This letter was uploaded as a separate file labeled 'Response to Reviewers'.

---

## [Decision Letter · Decision Letter 1]

22 Oct 2020

Factors associated with mechanical device-related complications in tube fed patients: A multicenter prospective cohort study

PONE-D-20-12166R1

Dear Dr. Gimenes,

We’re pleased to inform you that your manuscript has been judged scientifically suitable for publication and will be formally accepted for publication once it meets all outstanding technical requirements.

Kind regards,

Chun Chieh Yeh, M.D., Ph.D.

Academic Editor

PLOS ONE

Additional Editor Comments (optional):

After revisions, all comments from two reviewers were responded appropriately. Thus, we consider this article should be good for publication at its current content.

Reviewers' comments:

Reviewer's Responses to Questions

**Comments to the Author**

1. If the authors have adequately addressed your comments raised in a previous round of review and you feel that this manuscript is now acceptable for publication, you may indicate that here to bypass the “Comments to the Author” section, enter your conflict of interest statement in the “Confidential to Editor” section, and submit your "Accept" recommendation.

Reviewer #2: All comments have been addressed

2. Is the manuscript technically sound, and do the data support the conclusions?

Reviewer #2: Yes

3. Has the statistical analysis been performed appropriately and rigorously? 

Reviewer #2: Yes

4. Have the authors made all data underlying the findings in their manuscript fully available?

Reviewer #2: Yes

5. Is the manuscript presented in an intelligible fashion and written in standard English?

Reviewer #2: Yes

6. Review Comments to the Author

Reviewer #2: The authors have responded to all comments. The article is much improved and suitable to be published.

7. PLOS authors have the option to publish the peer review history of their article (what does this mean?). If published, this will include your full peer review and any attached files.

Reviewer #2: No

---

## [Editor Report · Acceptance letter]

3 Nov 2020

PONE-D-20-12166R1 

Factors associated with mechanical device-related complications in tube fed patients: A multicenter prospective cohort study 

Dear Dr. Gimenes:

I'm pleased to inform you that your manuscript has been deemed suitable for publication in PLOS ONE. Congratulations! Your manuscript is now with our production department. 

Kind regards, 

on behalf of

Dr. Chun Chieh Yeh 

Academic Editor

PLOS ONE